# Avoiding data loss: Synthetic MRIs generated from diffusion imaging can replace corrupted structural acquisitions for freesurfer-seeded tractography

Jeremy Beaumont[1,2]*, Giulio Gambarota[2], Marita Prior[3], Jurgen Fripp[1], Lee B. Reid[1]

1 The Australian e-Health Research Centre, CSIRO, Queensland, Australia, 2 Univ Rennes, INSERM, LTSI-UMR1099, Rennes, France, 3 Department of Medical Imaging, Royal Brisbane and Women's Hospital, Herston, Queensland, Australia

* jeremy.beaumont.2@gmail.com

**Data Availability Statement:** The code that supports the findings of this study is openly available in the FLAWS-Tools repository at https://github.com/jerbeaumont/FLAWS-Tools.git (https://doi.org/10.5281/zenodo.3993247). The sharing of

## Abstract

Magnetic Resonance Imaging (MRI) motion artefacts frequently complicate structural and diffusion MRI analyses. While diffusion imaging is easily 'scrubbed' of motion affected volumes, the same is not true for T1w or T2w 'structural' images. Structural images are critical to most diffusion-imaging pipelines thus their corruption can lead to disproportionate data loss. To enable diffusion-image processing when structural images are missing or have been corrupted, we propose a means by which synthetic structural images can be generated from diffusion MRI. This technique combines multi-tissue constrained spherical deconvolution, which is central to many existing diffusion analyses, with the Bloch equations that allow simulation of MRI intensities for given scanner parameters and magnetic resonance (MR) tissue properties. We applied this technique to 32 scans, including those acquired on different scanners, with different protocols and with pathology present. The resulting synthetic T1w and T2w images were visually convincing and exhibited similar tissue contrast to acquired structural images. These were also of sufficient quality to drive a Freesurfer-based tractographic analysis. In this analysis, probabilistic tractography connecting the thalamus to the primary sensorimotor cortex was delineated with Freesurfer, using either real or synthetic structural images. Tractography for real and synthetic conditions was largely identical in terms of both voxels encountered (Dice 0.88–0.95) and mean fractional anisotropy (intra-subject absolute difference 0.00–0.02). We provide executables for the proposed technique in the hope that these may aid the community in analysing datasets where structural image corruption is common, such as studies of children or cognitively impaired persons.

## 1 Introduction

Diffusion magnetic resonance imaging (MRI) is a form of medical imaging that can indirectly quantify certain aspects of tissue microstructure related to myelination [e.g. 1], axon density

patients' data from the Hospital dataset was not approved by the Royal Brisbane and Women's Hospital (RBWH) Human Research Ethics Committee. Therefore, the images from the Hospital dataset cannot be shared. For the HCP dataset, the FOD and T1 images (both real and synthetic) are available at https://data.csiro.au/collection/csiro:53349v2. All numerical results reported in this paper are detailed per subjects in Annexes S1-S5.

**Funding:** This research was partially supported by an Advance Queensland Research Fellowship (R-09964-01) and by the 'Region Bretagne' (France - ARED_BITRAST). The funders had no role in study design, data collection and analysis, decision to publish, or preparation of the manuscript.

**Competing interests:** The authors have declared that no competing interests exist.

[2], or cellular death. It also can, via tractography, delineate axonal pathways from which such microstructural measurements can be taken. These abilities make diffusion MRI a popular means of identifying and quantifying brain reorganisation and injury [3–5]. Studies utilising diffusion MRI commonly attempt to report measurements in specific white-matter tracts, or in specific regions of interest, which almost invariably requires identification of cortical or sub-cortical structures. Such parcellation typically relies upon an aligned structural MR image, such as a T1- or T2-weighted scan (referred to herein as 'structural images' for simplicity). This reliance is because most standard neuroimaging packages that can provide parcellations, such as Freesurfer [6], have been optimised for high-resolution images displaying particular tissue contrasts that are not apparent in diffusion images. The reliance on structural images to complete diffusion analyses can introduce four difficulties into analysis pipelines.

Firstly, cross-modality registration sometimes fails due to the meaningful difference between tissue contrasts combined with the typically low (2–2.5mm) spatial resolution of diffusion images [7]. Poor registration can be subtle, leading to biased measurements, or major, preventing analysis outright. Attempts have been made to tackle this issue through, for example, inverting the T1 contrast [8], registering T1 images to fractional anisotropy maps [9], or relying on mutual information as a registration cost function [10]. Such approaches only partially address contrast differences, however, and so cross-modal registration can still present as a major point-of-failure for fully-automated pipelines [9]. Multimodal registration can also prove computationally expensive and misregistrations can be time consuming to identify and correct.

Secondly, diffusion images typically display geometric distortions that are not found in structural images and can prevent perfect registration. The severity of such distortions depends on the scan parameters [11], and in some instances can be substantial. Correction of geometric distortion requires reverse-phase-encoded images or fieldmaps [11]. When such images are collected, correction often works reasonably well; when these are lost, corrupted, or difficult to use due to patient motion between scans, adequate registration is difficult or impossible to achieve. Although some specialised registration tools exist for such circumstances, the correction of eddy currents purely using registration can be problematic [11], and in the presence of larger deformations these tools can still leave several millimetres of misregistration between modalities [7].

Thirdly, reliance on structural MR images provides a greater risk that motion artefacts or other data loss will prevent analysis. In particular, diffusion MR sequences have a reasonable tolerance for motion as motion-affected volumes can be rejected from the 4D series and still leave sufficient information for analysis [12]. Conversely, the retrospective methods for correction of motion-corrupted structural images are usually based on the use of raw k-space data to obtain motion-robust image reconstructions [13–15], thus preventing their use for most studies in which the raw k-space data is not collected. The corruption of MR images with motion is particularly prevalent in young children and populations with brain injury, who are highly valuable participants but also highly likely to move during scanning [16] and often less tolerant to remaining in the scanner for repeat scans. Although less common, it is also possible that structural data can be corrupted, collected incorrectly, or simply lost. In such situations, the loss of structural images can completely preclude the analysis of the diffusion data using standard tools. As a concrete example, the constrained spherical deconvolution model [17] is commonly used for tractography and requires 45 uncorrupted diffusion encoding directions (assuming b = 3000s/mm$^2$) [18, 19]. This means 15 motion-affected volumes (25% of the acquired volumes) can be safely deleted from a classic 60-direction single-shell scan and the data remain analysable. We have previously reported on 123 scans of children and adolescents, the majority of whom had disability [20, 21]. Reviewing these data, 63% of scans demonstrated

motion artefacts in the diffusion scan but only 3.3% had 15 or more motion-affected volumes out of the 60 acquired. By contrast, 25% of structural scans displayed artefacts that precluded both structural and atlas-driven diffusion analyses [16].

Finally, diffusion acquisitions are typically limited to a lower spatial or angular resolution than desired due to scan time requirements. The requirement for high-quality structural images reduces the time available for diffusion acquisitions and thus their quality. Best practice is, where possible, to pre-allocate additional scan time for the re-acquisition of motion affected structural images, which further reduces available time for diffusion scans. This particularly affects studies of populations where motion artefacts are commonplace because substantial time must be allowed for re-acquisition of potentially several motion-affected images.

In this study we demonstrate a straightforward means of generating synthetic T1w and T2w images from a typical diffusion acquisition to allow for diffusion MRI analysis when *in-vivo* T1w and T2w images have been lost or motion corrupted. This process takes advantage of recent advances that allow approximate calculation of tissue compartments from both mul-tishell [22] and single shell [19] diffusion images. We demonstrate that by combining these modern methods with MRI simulation based on the use of the Bloch equations [23], synthetic images can be produced that are of sufficient quality to be used in place of genuine structural images in some standard diffusion tractography analyses. In many circumstances, this may eliminate the need to re-acquire motion-corrupted structural images or to reject participants from analysis due to poor quality structural images.

We are aware of three previous reports in this field. Roy et al. [24] proposed a means of generating T2w images, through multi-atlas registration and patch-matching, though this method relied on the collection of adequate-quality T1w images. More simply, Dhollander and Connelly [25] normalized a single diffusion-MR derived segmentation voxel-wise and multiplied it by experimentally derived values in order to generate an image qualitatively similar to a T1w image. Similarly, Cheng et al. [26] thresholded the b0 and a mean-diffusivity-like images to identify grey-matter- and white-matter-like tissues, then applied numerical constants to combine into a T1-like image. These latter approaches both generated T1-like images, but their utility is potentially limited by relying qualitative thresholding approaches to tissue segmentations, and a lack of a physical basis for conversion to the structural image. The present work extends beyond these works by demonstrating how one can utilise established segmentation approaches that require only diffusion data and perform genuine sequence simulation that allows arbitrary T1 or T2 contrasts to be generated. We demonstrate these techniques in both healthy individuals and those with sizeable pathology.

## 2 Methods

We first describe our proposed method to simulate structural MRIs using diffusion data, then describe *in-vivo* experiments designed to assess this method.

### 2.1 Proposed method

Our proposed method, summarised in Fig 1, requires that multi-tissue fibre orientation dispersion (FOD) maps have been calculated from diffusion MR data. These maps can be calculated from both single-shell and multi-shell diffusion acquisitions using standard tools without reliance on structural MR images. An example of this process is described in Section 3.2.2.

Once calculated, the white matter (WM), grey matter (GM) and cerebrospinal fluid (CSF) tissue components of the FOD maps are extracted and used to generate partial volume (PV)

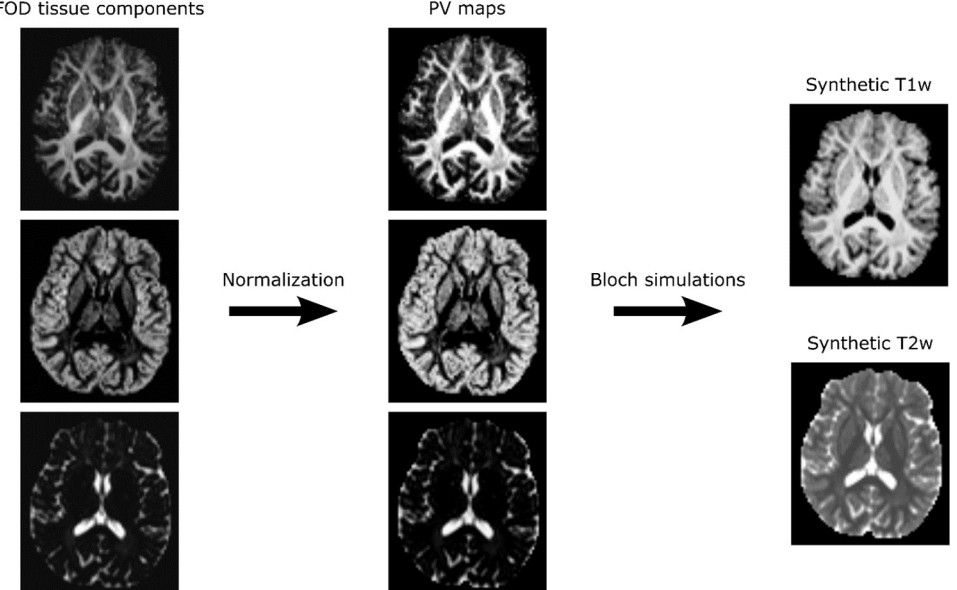

**Fig 1. Summary of the proposed MRI synthetization method.** After extracting the tissue components of the multi-tissue fibre orientation maps (FOD), partial volume (PV) maps are computed and used to generate synthetic T1w and T2w contrast with Bloch equations-based simulations.

maps as follows:

$$PV_A = \frac{TC_A}{TC_W + TC_G + TC_C} \tag{1}$$

Where $PV_A$ and $TC_A$ indicate the partial volume and the FOD tissue component of tissue $A$, respectively, and $TC_W$, $TC_G$ and $TC_C$ indicate the WM, GM and CSF tissue components of the FOD. The analytical solution of the Bloch equations for a selected sequence can then be applied to these PV maps to synthesize structural MR images. In the present study, we focus on T1-w and T2-w MRI signals which can be simulated using the following equations:

$$S_{T1w} = PV_W \ S_{mp}\left(\Theta_{mp}, \Phi_W\right) + PV_G \ S_{mp}\left(\Theta_{mp}, \Phi_G\right) + PV_C \ S_{mp}\left(\Theta_{mp}, \Phi_C\right) \tag{2}$$

$$S_{T2w} = PV_W \ S_{se}(\Theta_{se}, \Phi_W) + PV_G \ S_{se}(\Theta_{se}, \Phi_G) + PV_C \ S_{se}(\Theta_{se}, \Phi_C) \tag{3}$$

Where $S_{mp}$ and $S_{se}$ are the analytical solutions of the Bloch equations for the MPRAGE [27] and Spin Echo [28] sequences; $\Theta_{mp}$ and $\Theta_{se}$ represent the sequence parameters of the MPRAGE and spin echo sequences; and $\Phi_W$, $\Phi_G$ and $\Phi_C$ correspond to the magnetic properties of the WM, GM and CSF tissues, respectively.

Utilisation of Eqs 2 and 3 requires selection of scanning parameters and tissue magnetic properties. In the current study, the T1w scans were simulated with the ADNI MPRAGE sequence parameters ($\alpha = 9°$, $TE = 2.9 \ ms$, $TI = 900 \ ms$, $TR = 2300 \ ms$) [29]. Similarly, the T2w spin-echo scans were simulated with standard sequence parameters used in clinical imaging ($TE = 80 \ ms$, $TR = 4500 \ ms$) [30]. The tissue magnetic properties used to simulate the T1w and T2w signals were measured in previous studies conducted at 3T and are summarized in Table 1 [31–33].

**Table 1. Magnetic properties of white matter (WM), gray matter (GM) and cerebrospinal fluid (CSF) used for the T1w and T2w signal simulations.** These magnetic properties were measured in previous studies conducted at 3T [31–33].

|  | WM | GM | CSF |
|---|---|---|---|
| $T1$ (ms) | 830 | 1330 | 4000 |
| $T2$ (ms) | 80 | 110 | 1000 |
| $T2^*$ (ms) | 53 | 66 | 250 |
| $\rho$ [1] | 0.7 | 0.8 | 1.0 |

[1] The reported proton densities ($\rho$) are relative measures compared to the CSF proton density.

## 2.2 In-vivo experiments

We applied the proposed method to several diffusion datasets in order to assess the quality of the resulting images in terms of qualitative appearance, quantitative tissue contrast, and fitness for use as an integral component of a diffusion tractography pipeline.

**2.2.1 Input data.** Three datasets, named herein as the 'Human Connectome Project Multishell' (HCP-M), 'Human Connectome Project Single Shell' (HCP-S) and 'Hospital' datasets, were used in the current study. Scans in these datasets were mostly a convenient sample which had been preprocessed as part of a recent tractography study [9].

The HCP-M dataset included 10 healthy participants from the Human Connectome Project Young Adults 1200 Release [34]. Participants had been scanned twice on a Siemens Skyra Connectom scanner (Siemens Healthcare, Erlangen, Germany) with a 32-channel head coil. We used the preprocessed [35] T1w MPRAGE (1 $mm$ isotropic; $\alpha = 8°$; $TE = 2.14$ $ms$; $TI = 1000$ $ms$; $TR = 2400$ $ms$; acquisition time 7m 40s), T2w SPACE (1mm isotropic; $TE = 565$ $ms$; $TR = 3200$ $ms$; acquisition time 8m 24s) data, and preprocessed multishell diffusion data derived from a high quality acquisition (18 @ b = 0s/mm$^2$; 90 directions @ b = 1000s/mm$^2$; 90 directions @ b = 2000s/mm$^2$; 90 directions @ b = 3000s/mm$^2$; 1.25mm isotropic). For test-retest experiments (See Section 3.2.4), we utilised structural scans from two time points. For other experiments, only the first time point was used. Ethical consent was granted for use of HCP data.

The HCP-S dataset included the same participants as the HCP-M dataset, excepting that raw data were used (see below) for the diffusion data, which were downsampled and volumes were removed such that only a single shell remained (2 @ b = 0s/mm$^2$; 60 directions @ b = 3000s/mm$^2$; no directional repeats; 2mm isotropic). This dataset was designed to represent a modest single-shell HARDI acquisition often seen in the literature.

The Hospital dataset consisted of 12 neurosurgical patients scanned once on a Siemens Prisma scanner with a 64-channel head coil at the Herston Imaging Research Facility in Brisbane, Australia. Images were acquired as part of a clinical trial involving temporal lobe resection for epilepsy (n = 3 before surgery, n = 6 after surgery), glioma resection (n = 2 before surgery), or removal of arteriovenous malformations (n = 1 before surgery). This study included acquisition of a T1w MPRAGE scan (1mm isotropic, $\alpha = 9°$; $TE = 2.9$ $ms$; $TI = 900$ $ms$; $TR = 1900$ $ms$; acquisition time 4m 18s) and a modern multishell diffusion acquisition (12 @ b = 0s/mm$^2$; 20 directions @ b = 2000s/mm$^2$; 30 directions @ b = 1000s/mm$^2$; 60 directions @ b = 3000s/mm$^2$; 2mm isotropic; acquisition time 11m 30s) that is in use in a variety of studies [36–38]. Written informed consent was provided by all patients and the study approved by the Royal Brisbane and Women's Hospital (RBWH) Human Research Ethics Committee.

**2.2.2 Diffusion MRI processing.** Diffusion data from the HCP-S and Hospital datasets were processed automatically using the CONSULT neurosurgical planning pipeline [9]. This

pipeline prepared raw diffusion data for tractography using standard libraries and algorithms without requiring associated structural images. Steps included denoising via MRtrix 3.0 [39], removal of motion-corrupted volumes, brain mask calculation using MRtrix's dwi2mask, and eddy-current distortion correction using a combination of FSL's topup (http://fsl.fmrib.ox.ac.uk/fsl/fslwiki/TOPUP) and eddy_cuda 8.0. Subsequently, intensity inhomogeneities were corrected using N4-ITK [40]. A final brainmask was then recalculated using MRtrix's dwi2mask in conjunction with simple morphological operations.

Diffusion data from the HCP-M dataset were downloaded from the HCP server in their minimally preprocessed form. This preprocessing included correction for b0 intensity inhomogeneities, EPI distortion, eddy currents, head motion, gradient non-linearities, as well as reorientation and resampling to 1.25 mm isotropic. Brainmasks for these data were calculated in the same way as for HCP-S and Hospital datasets.

Tissue response functions were calculated using an unsupervised method [41] and FOD maps were calculated for white matter, grey matter, and cerebrospinal fluid using either multishell multitissue contrained spherical deconvolution (for multishell images) [22] or Single-Shell 3-Tissue constrained spherical deconvolution [19] (https://3Tissue.github.io; for HCP-S images). Fractional anisotropy (FA) maps were also calculated from the preprocessed data using MRtrix.

Synthetic T1w and T2w scans were generated from the resulting FOD maps, for all the three datasets, using the method described in Section 3.1.

**2.2.3 Synthetic and *in-vivo* structural image comparisons.** One standard means of assessing structural image quality is measuring intensity contrasts between brain tissues. Here, the brain-tissue contrasts of the *in-vivo* and synthetic structural images were measured using the following equation:

$$CN_{A/B} = \frac{S_A - S_B}{S_A + S_B} \tag{4}$$

Where $S_A$ and $S_B$ refer to the mean signal intensity of tissues $A$ and $B$, respectively. The contrast between brain tissues was measured in regions of interest (ROI) manually drawn in the corpus callosum for WM, caudate nucleus for GM and lateral ventricles for CSF. Each ROI comprised at least 20 contiguous voxels. To ensure that the contrast was comparable between scans with different resolutions, specific care was taken to avoid the inclusion of partial volume voxels within the ROIs.

The zero-normalized cross correlation was computed to further assess the similarity between the *in-vivo* and synthetic structural scans. This metric was preferred over standard similarity metrics, such as the structural similarity index, as it does not require that the images being compared have the same intensity range (T1w MPRAGE and T2w spin-echo contrasts are qualitative, and so repeat scans and synthetic images do not necessarily lie in the same intensity range as the first *in-vivo* image). To allow for the computation of the zero-normalized cross correlation, the synthetic images were spatially normalized to the *in-vivo* images using the affine registration tool provided by ANTS 2.1 [42]. The brain masks computed in the diffusion image space (see section 3.2.2) were then propagated in the structural image space to skull-strip the *in-vivo* images. The zero-normalized cross-correlation was computed as follows:

$$r = \frac{1}{N} \sum_{k \in \Omega} \frac{(I_k - \mu_I)(\hat{I}_k - \mu_{\hat{I}})}{\sigma_I \ \sigma_{\hat{I}}} \tag{5}$$

With $\Omega$ the set of voxels within the brain masks; $N$ the number of voxels within the brain mask; $I_k$ (respectively $\hat{I}_k$) the intensity of a given voxel $k$ in the *in-vivo* (respectively synthetic)

image; and $\mu_I$ and $\sigma_I$ (respectively $\mu_{\hat{I}}$ and $\sigma_{\hat{I}}$) the mean and standard deviation of the *in-vivo* (respectectively synthetic) brain tissue intensities.

## 2.3 Comparative performance in a diffusion tractography pipeline

We tested whether the synthetic images were of sufficiently high quality to be used in a standard Freesurfer-based diffusion tractography pipeline, described below. Similar to other tractography-generating processes, this pipeline required structural images for parcellation and diffusion images for tractography. For each subject, we ran this pipeline using (A) *in-vivo* structural scans acquired during the same scan session as the diffusion data (all datasets); (B) *in-vivo* structural scans acquired during a different scan session as the diffusion data ('*in-vivo* repeat'; HCP-M and HCP-S only); or (C) the synthetically generated structural scans (all datasets). The diffusion FOD image did not differ between each subject's acquisitions. To determine the influence of the structural scan on reproducibility of this pipeline, we compared these subjects' acquisitions in terms of both (1) spatial overlap of binarised tractography (Dice score) and (2) differences in FA and mean diffusivity (MD) sampled from this tractography. The former measure was to ascertain how synthetic structural images could impact the tractography itself. The latter was conducted to assess the practical significance of any error introduced by synthetic structural images, as the sampling of FA and MD are common use-cases for tractography.

All image processing was fully automated to avoid biasing results; no manual correction or process re-running was performed for any processing stage. Parcellation of structural images was performed using Freesurfer 6.0 [6]. Both the T1w and T2w images belonging to the HCP-M and HCP-S datasets were provided to Freesurfer. However, only the T1w images were utilised for the Hospital dataset because *in-vivo* T2w scans were not available.

When structural images were acquired *in-vivo*, the resulting parcellation was spatially normalized to the diffusion space by affinely registering the T1w scan to either the FA image (HCP-M and HCP-S) or mean 'b0' image (Hospital dataset) using ANTS 2.1 [42]. The mean 'b0' image was used for the Hospital dataset as it was already known to produce registrations of better quality for this particular dataset. When structural images were synthetic, no spatial normalization was required because image synthesis naturally creates images in native diffusion space.

The left superior thalamocortical tract was delineated using probabilistic tractography, using labels extracted from the Freesurfer parcellations. Specifically, the thalamus was used as the seeding ROI, while the left primary sensory cortex and left primary motor cortex were combined into a single inclusion ROI. Maximum streamline length was 80 *mm*. Streamlines were acquired using iFOD2 [43] until Tractography Bootstrapping stability criteria [44] were met to ensure the tractogram's reproducibility was not negatively affected by a low streamline count (minimum streamline count: 1000; min Dice: 0.975; reliability: 0.95; 1.25 *mm* isotropic; $t_{bin}$: $0.001 \times n$ *streamlines*). Other parameters were left at default values.

Subject-wise results were compared across runs. To assess overlap between tractograms, each tractogram was converted into a streamline density image at native diffusion resolution, thresholded at the Tractogram Bootstrapping stability threshold ($t_{bin}$), and binarised, in line with recommendations for when Tractogram Bootstrapping has been used [44]. The Dice scores between the binarised tractograms of each run were then calculated. To assess reproducibility of microstructural metrics, FA and MD were sampled from each tractogram using MRtrix's 'tcksample' [e.g. 4, 45, 46].

**2.3.1 Comparison of Freesurfer labels.** Our white-matter tract delineation was designed to indicate whether one could use synthetic images to perform a standard Freesurfer-reliant

tractography pipeline but we acknowledge that this does not utilise all Freesurfer-delineated regions. As such, we compared labels provided by Freesurfer using the generalised Dice similarity coefficient (gDSC) [47]. The gDSC is an adaption of the Dice Similarity Coefficient that can summarise overlap between identical labels when multiple labels are present. For each subject and pipeline we calculated one gDSC considering the 70 cortical grey matter labels, and another gDSC considering 16 deep grey matter labels. These were calculated as follows:

$$gDSC = 1 - 2 \ \frac{\sum_{l=1}^{N_l} \sum_n i_{ln} s_{ln}}{\sum_{l=1}^{N_l} \sum_n (i_{ln} + s_{ln})} \tag{6}$$

With $N_l$ the number of labels; $i_{ln}$ the value of voxel $n$ for label $l$ in the *in-vivo* image, $s_{ln}$ the value of voxel $n$ for label $l$ in the synthetic image. To ensure relevance to tractography, gDSC was calculated on parcellations in diffusion space. Furthermore, as tractography typically is seeded from or terminates at the grey-matter/white-matter interface, we restricted these labels to voxels in contact with the white matter (18-connectivity), as defined by the Freesurfer parcellation in question.

## 4. Results

### 4.1 Image quality and contrast

An example of PV maps generated from FOD tissue components is presented in Fig 1. Examples of synthetic T1w and T2w scans are presented in Fig 2. Qualitatively, all synthetic scans displayed a similar appearance to their corresponding *in vivo* scan, albeit at the resolution of their diffusion scan (1.25 or 2mm isotropic). Despite this limitation, the borders of individual gyri and subcortical structures were still easily identifiable in all images. Tissue contrast was qualitatively similar between *in-vivo* and synthetic images, despite small differences in acquisition and simulation protocols (Figs 1 and 2). Specifically, the synthetic T1w scans were characterized by a good suppression of the CSF signal and by a high contrast between grey matter, white matter, and CSF, as is typical for MPRAGE acquisitions. Similarly the synthetic T2w scans displayed comparable tissue contrasts to those of the *in-vivo* T2w scans, including the typical hyper-intensity of the CSF signal. Synthetic images derived from single and multi-shell diffusion scans produced similar tissue contrast, despite relying on meaningfully different diffusion FOD algorithms and resolutions. Notably, images were also realistic for patients with tumours and post-surgical cavities.

The quantitative brain tissue contrast measurements obtained for both T1w and T2w *synthetic* images were close to their *in-vivo* counterparts for the HCP-M, HCP-S and Hospital datasets (Tables 2 and 3). The brain tissue contrasts were not significantly different between *in-vivo* and *synthetic* images (Tables 2 and 3), except for the T1w WM/GM contrast in the Hospital dataset (Bonferroni-corrected Wilcoxon signed rank test, p = 0.02) and the WM/CSF and GM/CSF contrasts in the HCP-M dataset (Bonferroni-corrected Wilcoxon signed rank tests, both p = 0.04). The contrast difference between *in-vivo* and *synthetic* images remained very small for these cases (Tables 2 and 3).

A very strong zero-normalized cross-correlation was measured between the co-registered T1w *in-vivo* and synthetic images for the Hospital (mean: 0.87; range: 0.82–0.90), HCP-M (mean: 0.86; range: 0.84–0.87) and HCP-S (mean: 0.86; range: 0.83–0.88) datasets. The zero-normalized cross-correlation measured between the co-registered T2w *in-vivo* and synthetic images was strong for both HCP-M (mean: 0.75; range: 0.67–0.80) and HCP-S (mean: 0.68; range: 0.64–0.73) datasets.

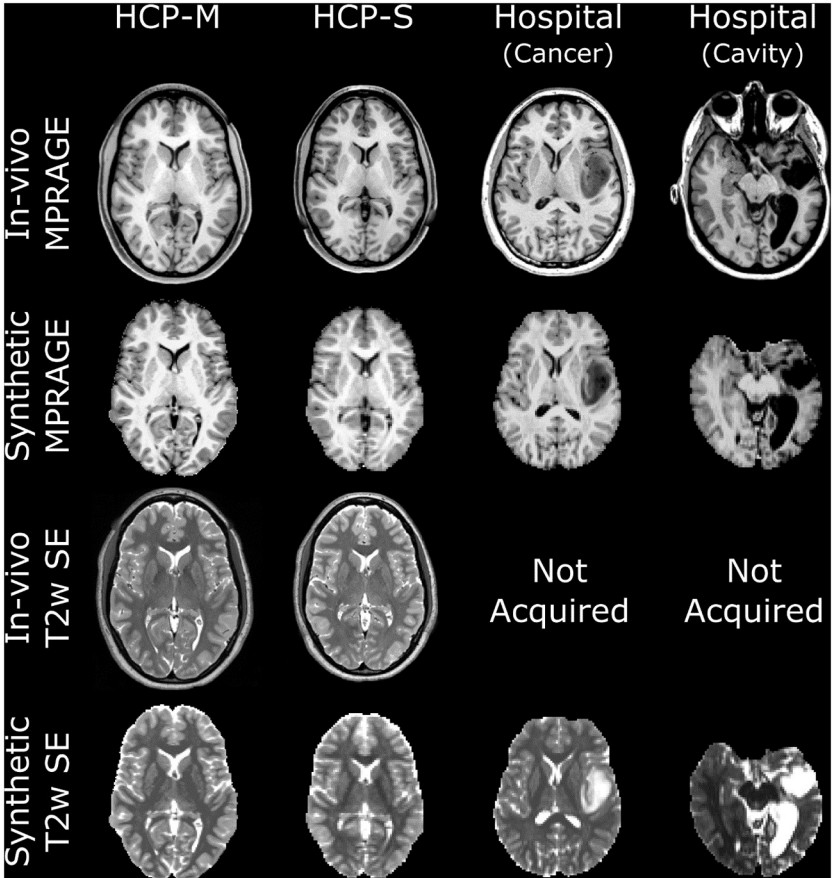

**Fig 2.** In-vivo and synthetic T1w and T2w images obtained from the HCP-M (far left), HCP-S (middle left) and Hospital (middle and far right) datasets. The synthetic T1w images display contrasts qualitatively similar to that of the in-vivo scans for all datasets, even in presence of tumours (middle right) and post-surgical cavities (far right). The T2w contrast was similar between the synthetic and in-vivo scans for the HCP-M and HCP datasets, including the expected hyper-intensity of the cerebrospinal fluid signal. For the Hospital dataset, the in-vivo T2w images were not acquired but synthetic images provided similar contrast to that which could be expected from an in-vivo scan, including in the areas of pathology.

**Table 2. Mean brain tissue contrast measured for in-vivo and synthetic T1w scans on the Hospital, HCP-M and HCP-S datasets.**

|  | WM/GM | | | WM/CSF | | | GM/CSF | | |
|---|---|---|---|---|---|---|---|---|---|
|  | Mean | Range | p-value | Mean | Range | p-value | Mean | Range | p-value |
| *In-vivo* Hospital | 0.14 | 0.12–0.15 | N/A | 0.59 | 0.56–0.65 | N/A | 0.49 | 0.46–0.55 | N/A |
| *In-vivo* HCP | 0.12 | 0.09–0.15 | N/A | 0.56 | 0.54–0.59 | N/A | 0.47 | 0.45–0.49 | N/A |
| Synthetic Hospital | 0.09 | 0.07–0.12 | 0.02 | 0.56 | 0.52–0.60 | 0.30 | 0.49 | 0.47–0.52 | 1.00 |
| Synthetic HCP-M | 0.10 | 0.07–0.12 | 0.10 | 0.50 | 0.39–0.60 | 0.17 | 0.43 | 0.31–0.55 | 0.34 |
| Synthetic HCP-S | 0.13 | 0.10–0.15 | 1.00 | 0.50 | 0.42–0.58 | 0.07 | 0.40 | 0.33–0.48 | 0.05 |

HCP datasets used identical in-vivo structural images and so appear together in one row. P-values indicate comparisons of in-vivo vs synthetic contrasts using Bonferroni-corrected Wilcoxon signed rank tests. The contrasts measured for the synthetic scans are similar to the contrasts measured for the in-vivo scans, despite slight differences between the imaging protocols.

**Table 3. Mean brain tissue contrast measured for in-vivo and synthetic T2w scans on the Hospital, HCP-M and HCP-S datasets.**

|  | WM/GM | | | WM/CSF | | | GM/CSF | | |
|---|---|---|---|---|---|---|---|---|---|
|  | Mean | Range | p-value | Mean | Range | p-value | Mean | Range | p-value |
| *In-vivo* HCP | 0.16 | 0.12–0.21 | N/A | 0.53 | 0.50–0.57 | N/A | 0.41 | 0.37–0.44 | N/A |
| Synthetic Hospital | 0.15 | 0.11–0.19 | N/A | 0.58 | 0.56–0.59 | N/A | 0.47 | 0.44–0.48 | N/A |
| Synthetic HCP-M | 0.16 | 0.13–0.18 | 1.00 | 0.58 | 0.55–0.59 | 0.04 | 0.46 | 0.42–0.49 | 0.04 |
| Synthetic HCP-S | 0.19 | 0.16–0.23 | 0.17 | 0.55 | 0.51–0.58 | 0.92 | 0.40 | 0.36–0.44 | 1.00 |

P-values indicate comparisons of in-vivo vs synthetic contrasts using Bonferroni-corrected Wilcoxon signed rank tests. P-values for the Hospital dataset are unavailable as no in-vivo T2w scan was acquired for this dataset. The synthetic HCP-M dataset WM/CSF and GM/CSF contrasts are significantly higher than their in-vivo counterparts. Although the absolute WM/CSF and GM/CSF contrast difference remained very small. The other synthetic contrasts were very close to their in-vivo counterparts, despite the use of different magnetic resonance sequences to acquire and simulate the signals.

## 4.2 Comparative performance in a diffusion tractography pipeline

Freesurfer succeeded for all but one participant. This participant, from the Hospital dataset, presented with a glioma in the left temporal lobe, preventing adequate Freesurfer performance when provided with either *in-vivo* or synthetic input data (Fig 2, middle-left column). A qualitative assessment indicated that Freesurfer parcellations obtained from synthetic datasets were similar to those obtained from *in-vivo* data, as shown in Fig 3. Notably, near the white matter/grey matter interface, from which tractography is typically seeded, Freesurfer parcellations were qualitatively similar between corresponding synthetic and *in vivo* images. However, the synthetic images at 2mm resolution often produced inaccurate segmentation of the pial surface by Freesurfer (Fig 3, S1 Table in S1 File).

Tractography was successfully performed for all the datasets for which Freesurfer parcellation did not fail. Tractography was qualitatively similar between in-*vivo*- and synthetic-structural pipelines (Fig 4). Median Dice coefficients for binarised tractography between the *in-vivo*- and synthetic-structural pipelines were 0.93 (HCP-M), 0.93 (HCP-S), and 0.90 (Hospital) with respective worst-case performances of 0.90, 0.90, and 0.88 (Fig 5). Structural scans were available from a second time point for all HCP datasets. The tractography pipeline was repeated using the same diffusion data but with these alternative *in-vivo* structural scans. These *in-vivo* vs *in-vivo-repeat* comparisons demonstrated similar median Dice coefficients to, and poorer worst-case performance than, the aforementioned *in-vivo* vs synthetic comparisons (Fig 5). Numerical differences between the *in-vivo-repeat* and the *synthetic* Dice coefficients were not statistically significant for HCP-S or HCP-M datasets (Bonferroni-corrected Wilcoxon Signed-Rank Tests, both p>0.1; see Fig 5). The poorer worst case performance of the *in-vivo-repeat* analysis may be explained by variations in the quality of registration between structural and diffusion scans: a step not required by the synthetic pipeline which guarantees perfect alignment between modalities.

FA values sampled from tractography did not differ significantly between pipelines for the HCP-M or HCP-S datasets (Bonferroni-corrected Wilcoxon Signed-Rank tests versus the *in-vivo*-structural pipeline FA values: HCP-M Synthetic, p = 1; HCP-M *in-vivo* repeat, p = 1; HCP-S Synthetic, p = 0.11; HCP-S *in-vivo* repeat 0.37).

Although a statistically significant difference was found between *in-vivo* and synthetic pipelines for the Hospital dataset (p = 0.02), this median difference was only 0.01, or 2.2%, which is well below both inter-group and longitudinal differences typically reported in imaging studies [4, 45, 46].

MD values sampled from tractography did not differ significantly between pipelines for the HCP-S dataset (Bonferroni-corrected Wilcoxon Signed-Rank tests versus the *in-vivo*-

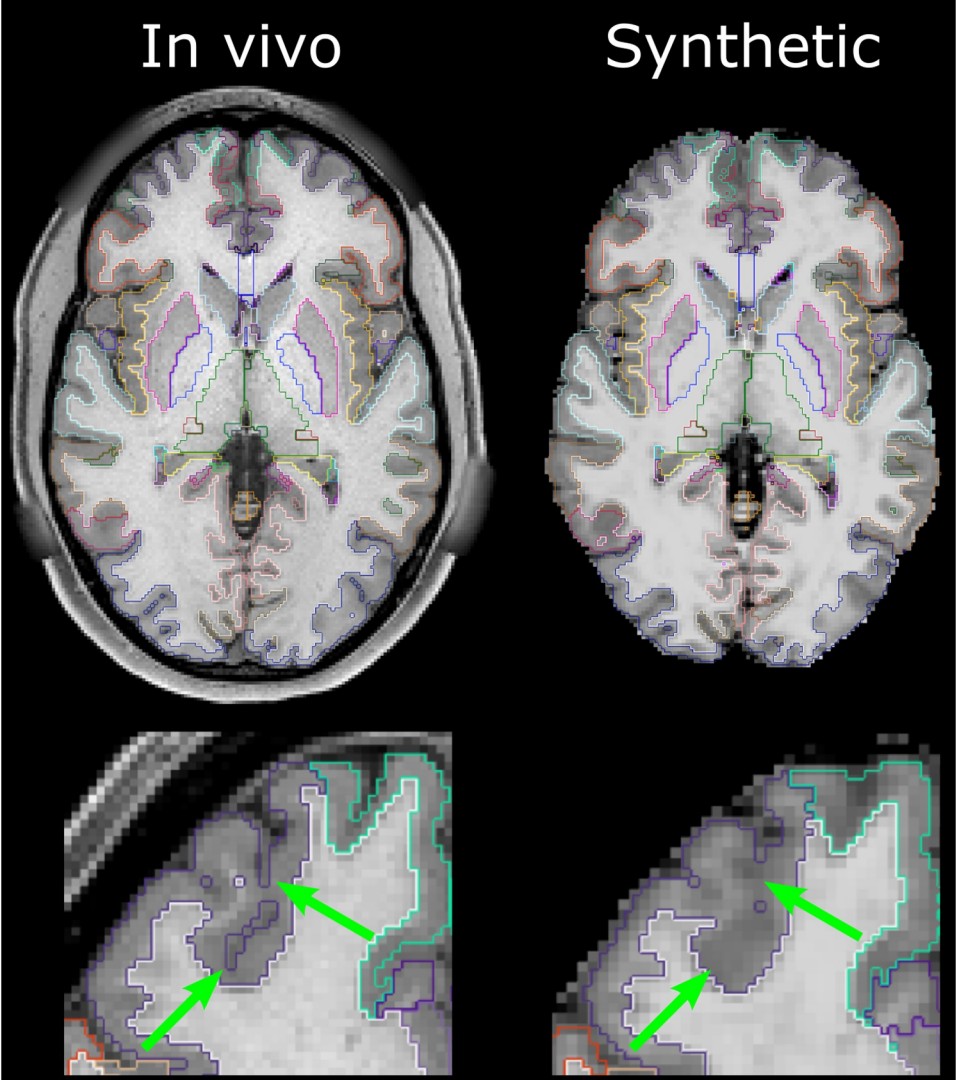

**Fig 3. Example of freesurfer parcellations obtained from the in-vivo (left) and synthetic (right) scans of the Hospital dataset.** The bottom row shows a magnified section of the frontal lobe. A qualitative assessment indicated that the synthetic parcellations were similar to the in-vivo parcellations in subcortical structures and near the grey-matter/white-matter interface from which tractography is typically seeded. However, the lower resolution provided by the synthetic data (2 mm isotropic) reduced the accuracy of the pial surface segmentation, as highlighted by the green arrows.

structural pipeline MD values: HCP-S Synthetic, p = 1; HCP-S *in-vivo* repeat, p = 1). In other datasets, all other comparisons with the first *in-vivo* pipeline showed differences that were significant, or bordered on significance (HCP-M *in-vivo* repeat, p = 0.01; HCP-M synthetic, p = 0.06; Hospital synthetic, p = 0.08), but these median differences (0.33%, 0.24%, and 0.78% respectively) were again well below inter-group and longitudinal differences of clinical or scientific significance [E.g. 4, 48].

## 4.3 Comparison of Freesurfer labels

Table 4 shows generalised Dice similarity coefficients for 70 cortical grey matter labels, or 16 deep grey-matter labels. These gDSCs reflect similarities between label obtained from the first

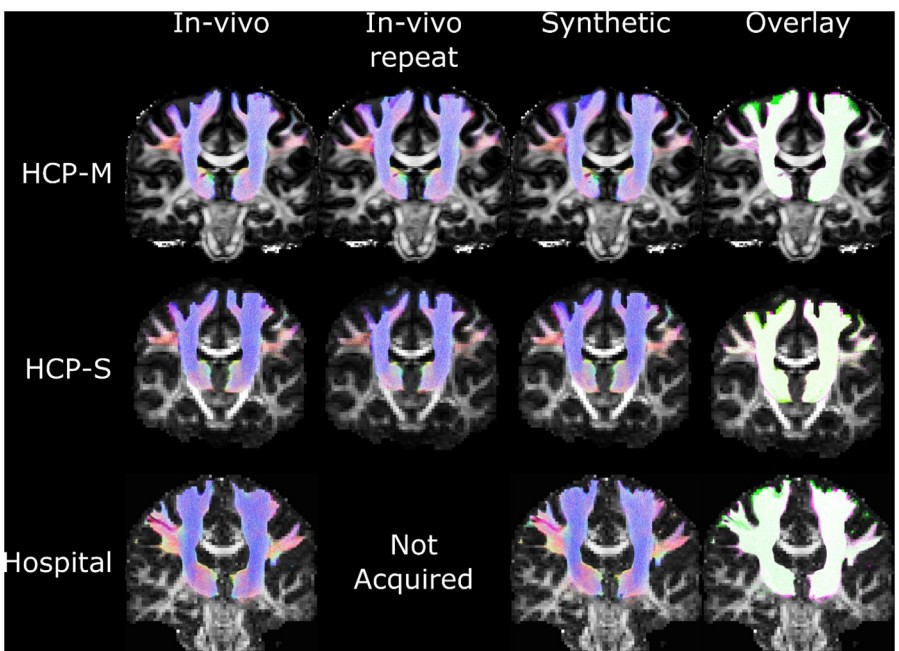

**Fig 4. Example tractography from the in-vivo and synthetic pipelines.** The rightmost column shows overlap of the in-vivo tractography (from the leftmost column; green) with the synthetic tractography (fuchsia; white indicating overlap). Tractography densities showed clear correspondences between in vivo and synthetic pipelines for HCP-S, Hospital, and HCP-M datasets. The datasets shown here were selected at random.

*in-vivo* structural scan and labels obtained using either the repeat *in-vivo* structural scan or synthetic structural image. For the HCP-M dataset, median gDSCs were typically lower for synthetic images than repeat *in-vivo* data (p = 0.01 for both comparisons using uncorrected Wilcoxon Signed-Rank tests). The worst-case performances of *in-vivo* and synthetic pipelines were similar on this dataset. For the HCP-S dataset, differences between *in-vivo* and synthetic pipeline gDSCs were not statistically significant (p = 0.65 for cortical GM and p = 0.51 for deep GM using uncorrected Wilcoxon Signed-Rank tests). On the HCP-S dataset, the synthetic pipeline demonstrated considerably better worst-case performance than the *in-vivo* pipeline. Dice coefficients for Freesurfer-delineated tissues can be found in S1 Table in S1 File.

## 5 Discussion

Motion artefacts are commonplace in MRI acquisition, particularly with children and cognitively impaired persons [16, 49]. Diffusion imaging can be easily 'scrubbed' of motion affected volumes, typically leaving usable data [18, 19]. However, the same is not true for structural images which are critical to most diffusion analyses. As such, motion or other corruption of structural images can lead to disproportionate data loss [16, 49]. To alleviate this issue, we have proposed a means by which synthetic structural images can be generated from diffusion MR images. We are aware of one previous report [25] in which a single diffusion-MR derived segmentation was normalised voxel-wise and multiplied by experimentally derived values in order to generate an image qualitatively similar to a T1w image. Similarly, Cheng et al. [26] applied thresholding to processed diffusion weighting images to identify grey-matter- and white-matter-like tissues, then applied constants to combine into a T1-like image. These latter approaches both generated T1-like images, but their utility is potentially limited by relying

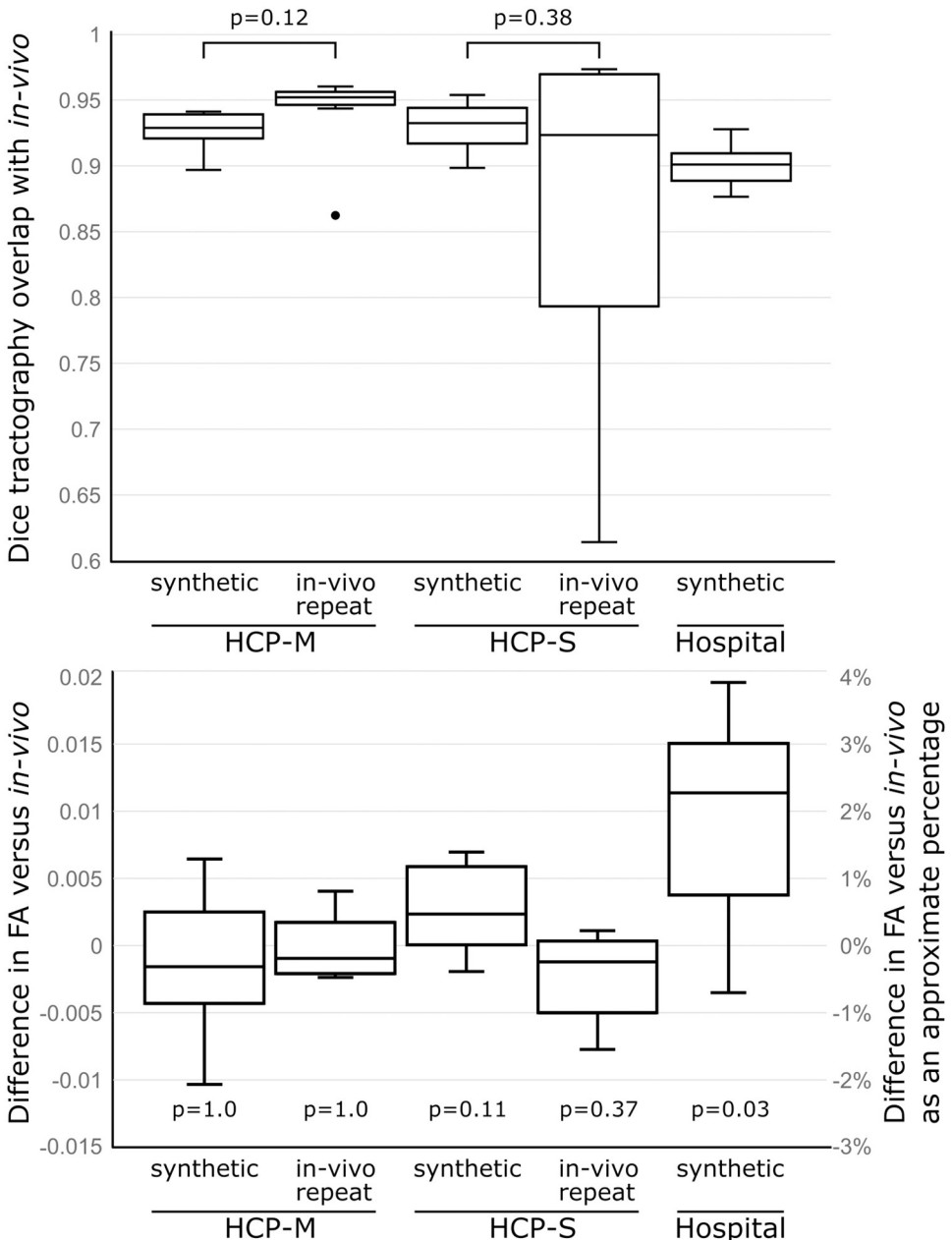

**Fig 5. The tractography pipeline was run first using genuine ('in-vivo') diffusion and structural MR images, and then run again using the same diffusion data but with either a synthetic structural image ('synthetic') or a structural scan acquired during another scanning session ('in-vivo repeat').** Top: Dice scores for overlap of tractography between runs using in-vivo data and either synthetic or in-vivo repeat data. Median dice scores did not differ significantly between synthetic and in-vivo repeat runs after correction for multiple comparisons. Bottom: Mean tract FA values sampled from the tractography, compared with the in-vivo tractography. Differences between the in-vivo versus synthetic or in-vivo-repeat runs were below the level of practical significance. Approximate percentages were calculated by dividing differences by the mean FA of all datasets (0.52). Displayed p-values are corrected for multiple comparisons.

**Table 4. Generalised Dice coefficients for 70 cortical grey matter labels (Cortical GM) or 16 deep grey-matter labels (Deep GM).**

|  | Cortical GM | Deep GM |
|---|---|---|
| Synthetic Hospital | 0.56 (0.47–0.63) | 0.59 (0.57–0.63) |
| *In-vivo* Repeat HCP-M | 0.75 (0.34–0.76) | 0.70 (0.41–0.75) |
| Synthetic HCP-M | 0.59 (0.34–0.63) | 0.55 (0.40–0.62) |
| *In-vivo* Repeat HCP-S | 0.68 (0.17–0.90) | 0.74 (0.07–0.93) |
| Synthetic HCP-S | 0.60 (0.52–0.64) | 0.62 (0.54–0.69) |
|  | *In-vivo* Repeat vs Synthetic Dice Scores | |
| HCP-M | p = 0.005 | p = 0.005 |
| HCP-S | p = 0.65 | p = 0.51 |

Median values are shown; ranges appear in brackets. Dice coefficients are a comparison between the listed scan and the first in-vivo scan and restricted to voxels at the GM/WM interface in diffusion space. P-values indicate comparison of dice coefficients between in-vivo Repeat vs Synthetic scans using Wilcoxon signed rank tests.

qualitative thresholding approaches to tissue segmentations, and a lack of a physical basis for conversion to the structural image. We have expanded on these ideas by using a well-established physical model that allows precise scanning parameters to be simulated; this is demonstrated here by the synthesis of specific T1w MPRAGE and T2w spin-echo sequences from both single-shell (HCP-S dataset) and multi-shell (HCP-M dataset, Hospital dataset) diffusion scans. We demonstrated that the synthetic images generated for these 32 participants, including those displaying pathology, were both visually convincing and had tissue contrasts in line with images acquired *in-vivo*. In addition, the strong zero-normalized cross-correlations measured between the synthetic and *in-vivo* images further highlighted their structural similarities.

To demonstrate the utility of the proposed method, we performed Freesurfer-driven tractography for 32 participants. Freesurfer is a well-established program and is regularly relied on for diffusion analyses but can, like many packages, fail if low-quality images are provided. Here, Freesurfer was able to segment brains successfully in all instances for both *in-vivo* and synthetic data excepting for one subject presenting with a glioma for which it failed with both real and synthetic images. Due to the lower resolution of the synthetic data, the pial surface was not always as well defined in the synthetic-image parcellation as in the *in-vivo* image parcellation. However, this is unlikely to be of serious concern for tractography-based studies which are more reliant on an accurate grey-matter/white-matter interface, which was generally of reasonable quality (Fig 3), particularly for the 2mm HCP-S dataset (Table 4). Indeed, tractography derived from *in-vivo* structural and synthetic-structural parcellations overlapped well (median Dice ≥ 0.90) for all three types of diffusion acquisition tested in this study. A common use for tractography is to identify a tract from which microstructural measures, such as FA or MD, can be sampled. Here, FA and MD sampled from tractography of the *in-vivo* and synthetic-image pipelines differed well below the level of practical significance for most forms of analysis (Fig 5).

An alternative to the proposed method, available only to longitudinal diffusion MRI studies, is to replace a corrupted structural image with another acquired at another time point. We tested the efficacy of this alternative for the HCP-S and HCP-M datasets by running the Freesurfer-based pipeline with T1w and T2w scans acquired at a second time point. In general, this did not produce meaningfully better results, in terms of tractography Dice scores, MD, or FA, than using a synthetic image. Visual inspection suggested that performance of this *in-vivo*-repeat run sometimes suffered from slight mis-registrations between the real structural images

and diffusion images, reminding that the negative impact of the lower- resolution synthetic images is somewhat counter balanced by their guaranteed perfect alignment to the diffusion image.

We also calculated generalised Dice coefficients (gDSC) for 70 cortical GM labels and 16 deep GM labels, at the GM/WM interface (Table 4). For the HCP-S dataset, gDSC scores for the *synthetic* and *in-vivo* repeat pipelines were comparable. Results for the HCP-M dataset, however, implied that groups who work with high resolution (1.25mm) diffusion data whose pipelines rely on a particularly accurate cortical parcellation would do best to rely on an *in-vivo* structural image (if available), rather than a synthetic structural image. The degree to which the differing parcellation accuracies will affect tractography depends on the specifics of the tractography. For example, when we performed tractography of the corticothalamic tracts in this study, we used cortical and thalamic ROIs from Freesurfer. The ROIs produced by the *in-vivo* pipeline had higher gDSC scores (median gDSCs of 0.73 for cortical GM, 0.64 for thalamus) than the synthetic pipeline (0.58 cortical GM, 0.48 thalamus). Nevertheless, these differences did not meaningfully affect the physical location of the tractography or the microstructural measurements taken from it (Fig 5). This may not be the case in all situations, but it serves as a reminder that subtle changes in seed or termination ROIs do not necessarily bias diffusion measurements. Notably, the worst-case performance was equal between the synthetic and in-vivo pipelines for high-resolution (HCP-M) diffusion data, while at the more typical resolution of 2mm (HCP-S), the synthetic pipeline had better worst-case performance than the *in-vivo* pipeline. Such results highlight that while a genuine structural image is usually preferable, when this is lost or corrupted a synthetic image can act as a worthwhile substitute.

One strength to our method is a reliance on a well-established physical model that makes a relatively small number of assumptions. While hand-tuned or machine-learning based approaches could in principle generate similar outputs, these are inherently tuned to their training data, which sometimes leaves general use uncertain–for example, when presented with lesions, low quality data, or high resolution inputs. By contrast, our proposed method is resolution independent and, as demonstrated here, was able to generate convincing images from data acquired on two scanners, using two different tissue response function methods, and three different acquisition parameters. It also was able to produce convincing and usable images where pathology and post-surgical cavities were present without requiring tuning of any kind. One limitation of the current approach is the assumption that the brain tissues have the same relaxometry values for all subjects, while the subject relaxometry values may change due to the subject age or brain disease. However, the relaxometry values are used as input variables in the solutions of the Bloch equations for the synthetization of the structural images, thus allowing to easily tailor the model by using relaxometry values corresponding to the population of interest (ie. young vs aged subjects, type of pathology, . . .). Nevertheless, the proposed method is likely to not work for neonates as it models tissues that have not yet fully developed in the neonate brain.

Our proposed method is dependent on the 'single-shell 3-tissue' (SS3T) or 'multishell-multitissue' (MSMT) diffusion models, which in turn are reliant on an adequate number of diffusion directions being available. We have not explored the effects of progressively reducing the number of directions to simulate movement because the requirements of SS3T and MSMT have already been explored [18, 19, 22]. Furthermore, our method has no additional requirements over those of typical tractography that uses constrained spherical deconvolution (at least 45 directions at a b-value of 3000s/mm$^2$). If these requirements are not met high-quality tractography will not be possible, regardless as to whether an *in-vivo* or synthetic structural is used. Importantly, the proposed method is intended to enable diffusion MRI analyses of white matter; it is not intended to produce images for clinical interpretation nor for other types of

analyses that would normally rely on precise tissue-contrast for certain pathologies or sharp pial surfaces. For this intended use-case, the major limitation of the proposed method is that the synthetic images produced are at the resolution of the diffusion data. Although it is possible to apply up-sampling and super-resolution techniques, we have not attempted to do so here because the synthetic images were of sufficient quality to obtain adequate quality parcellations and diffusion metrics. It is also worth considering that the proposed method provides some insurance against motion-corrupted structural images, therefore, scan time does not necessarily have to be pre-allocated for potential re-acquisition of motion-affected structural scans. This, in turn, may provide some studies with additional scan time that can potentially be used to modestly improve the resolution of their diffusion data.

In conclusion, we have presented a simple and physically constrained means of synthesising structural images with customisable acquisition parameters from diffusion MRI. These images are of adequate quality to be used with standard parcellation tools, allowing analysis of diffusion data that would otherwise be impossible due to motion-corrupted or non-acquired scans.

## Supporting information

**S1 File.**
(ZIP)

## Acknowledgments

Data were provided in part by the Human Connectome Project, WU-Minn Consortium (Principal Investigators: David Van Essen and Kamil Ugurbil; 1U54MH091657) funded by the 16 NIH Institutes and Centers that support the NIH Blueprint for Neuroscience Research; and by the McDonnell Center for Systems Neuroscience at Washington University. The authors are grateful to the Dr Katie McMahon, Ms Clare Berry, and the team at the Herston Imaging Research Facility for their generous scan time allowances, and technical support acquiring some of the data presented here. The authors acknowledge the facilities of the National Imaging Facility, a National Collaborative Research Infrastructure Strategy (NCRIS) capability, at the Herston Imaging Research Facility.

## Author Contributions

**Conceptualization:** Jeremy Beaumont, Lee B. Reid.

**Formal analysis:** Jeremy Beaumont, Lee B. Reid.

**Funding acquisition:** Giulio Gambarota, Marita Prior, Jurgen Fripp, Lee B. Reid.

**Investigation:** Jeremy Beaumont, Lee B. Reid.

**Methodology:** Jeremy Beaumont, Lee B. Reid.

**Project administration:** Marita Prior, Jurgen Fripp, Lee B. Reid.

**Software:** Jeremy Beaumont, Lee B. Reid.

**Supervision:** Giulio Gambarota, Jurgen Fripp, Lee B. Reid.

**Validation:** Jeremy Beaumont, Lee B. Reid.

**Visualization:** Jeremy Beaumont, Lee B. Reid.

**Writing – original draft:** Jeremy Beaumont, Lee B. Reid.

**Writing – review & editing:** Jeremy Beaumont, Giulio Gambarota, Marita Prior, Jurgen Fripp, Lee B. Reid.

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
