## [Decision Letter · Decision Letter 0]

11 Jun 2021

PONE-D-21-03790

Avoiding Data Loss: Synthetic MRIs Generated from Diffusion Imaging Can Replace Corrupted Structural Acquisitions For Freesurfer-Seeded Tractography

PLOS ONE

Dear Dr. Beaumont,

Thank you for submitting your manuscript to PLOS ONE. After careful consideration, we feel that it has merit but does not fully meet PLOS ONE’s publication criteria as it currently stands. Therefore, we invite you to submit a revised version of the manuscript that addresses the points raised during the review process.  In particular, all reviewers raised concerns about the need for quantitative data in the evaluation of the method.  In addition, it was suggested that several relevant papers should be cited and discussed.  

We look forward to receiving your revised manuscript.

Kind regards,

Dzung Pham

Academic Editor

PLOS ONE

Journal Requirements:

2. Thank you for stating the following in the Competing Interests/Financial Disclosure * (delete as necessary) section:

This research was partially supported by an Advance Queensland Research Fellowship (R-09964-01) and by the ‘Region Bretagne’ (France - ARED_BITRAST)

We note that you received funding from a commercial source: Advance Queensland Research Fellowship

Please include your amended Competing Interests Statement within your cover letter. We will change the online submission form on your behalf."

Reviewers' comments:

Reviewer's Responses to Questions

**Comments to the Author**

1. Is the manuscript technically sound, and do the data support the conclusions?

Reviewer #1: Yes

Reviewer #2: Yes

Reviewer #3: Partly

2. Has the statistical analysis been performed appropriately and rigorously? 

Reviewer #1: Yes

Reviewer #2: Yes

Reviewer #3: No

3. Have the authors made all data underlying the findings in their manuscript fully available?

Reviewer #1: Yes

Reviewer #2: Yes

Reviewer #3: Yes

4. Is the manuscript presented in an intelligible fashion and written in standard English?

Reviewer #1: Yes

Reviewer #2: Yes

Reviewer #3: Yes

5. Review Comments to the Author

Reviewer #1: The authors have presented and demonstrated a method to synthesize structural MRI data which would allow for increased use of scanning time for diffusion MRI data particularly in populations who are prone to movement during the scan.

1. One of the potential advantages of this method is that it could be used in a scenario with participants who move during the scan. Presumably there would be motion during the diffusion scan which could be corrected during processing. In the datasets presented in the paper how many volumes had to be excluded due to motion? Does motion affect the ability to generate the synthesized structural image and if so, to what extent?

2. Were there comparisons of fiber tract orientation or other tensor based metrics in addition to using FA as a quantitative metric to assess the quality of tractography using a synthetic structural image?

A relevant paper to cite is by Hu Cheng et al (June 2020) titled "Segmentation of the brain using direction-averaged signal of DWI images. Adding this to the discussion would place the proposed method in context of current alternative approaches.

Reviewer #2: This paper presents a physics based method for synthesizing T1w and T2w MR images from diffusion MR images, which first estimates tissue volume from diffusion data using multi-tissue spherical deconvolution, and then uses the estimated tissue volume for bloch simulation of different sequences such as MPRAGE. The method uses the synthesized T1w data for FreeSurfer reconstruction for brain segmentation, which is then used for delineating brain ROIs for diffusion tractography. This method is novel and interesting, and might be a potential useful tool for the neuroimaging community. The paper is clearly written, except for the intro which needs better motivation and literature review. The experiments are well designed but the evaluation of the proposed method should be more comprehensive and thorough. My detail comments are as below.

(1) The introduction for diffusion MRI and structural MRI in the first paragraph is not very friendly to non-MRI readers. Maybe provide more background. Also maybe provide more explanations on why “such analyses almost always require identification of cortical or subcortical structures using an aligned structural MR image, such as a T1- or T2-weighted scan”.

(2) The first paragraph to some extent provides some motivation on why “Such reliance on structural images can introduce four 4 difficulties into analysis pipelines”, which need some refinement and detailed explanation. The main issue between the co-reg between the T1w and the diffusion data is the susceptibility induced non-linear distortion present in the diffusion data. This non-linear distortion can be corrected e.g. using blip-up/blip-down b=0, but this data might be not acquired in some cases. The other main reason is that sometimes the T1w is also missing. Can refer to these papers: Little and Beaulieu. NeuroImage 2021, 237, 118105. Bhushan, Haldar, et al. NeuroImage 2015, 115, 269-280.

(3) In general, the intro is too brief which requires more details. Also the intro needs to better discuss previous similar works and do a better literature review.

(4) The paper only qualitatively compares the synthesized images and the actually acquired T1w or T1w images. Maybe consider comparing them quantitatively using metrics such as SSIM using up-sampled diffusion synthesized images.

(5) Even though the brain segmentation is the main focus of the method, there is no evaluation or comparison of the FreeSurfer results from synthesized images and actual images, which is much easier to perform than for images. Figure 3 only shows tissue and cortical segmentation visually. The Dice coefficients or some overlapping metrics for the segmentation of each cortical area and each brain region from synthesize and actual images should be reported to quantify the performance of the proposed method. The segmentation from the diffusion synthesized images can be upsampled using nearest neighbour resampling.

(6) Showing and comparing tractography results on only one tract, i.e., superior thalamocortical tract, is not sufficient. Results from more tracts need to be reported and ideally the whole-brain structural connectivity should be reconstructed and compared. The reconstruction of the structural connectivity requires the use of each segmented brain region from the synthesized images, and therefore could more accurately and comprehensively reflect the quality of the synthesized images and their FreeSurfer reconstruction results.

Reviewer #3: Authors present an approach for generating synthetic T1w and T2w image from diffusion MRI images by using Bloch equations along with tissue partial volume estimates from diffusion MRI. The synthetic images are used in a diffusion tractography pipeline. The method is clearly described and results are presented on HCP and a private hospital data. In my opinion, the results do not provide evidence of robustness to nuisances and repetition of diffusion MRI acquisition. Also, the method assumes very simple tissue model parameter (table 1) that is assumed to be constant across population and throughout the brain, which will most likely be not true in scenarios "where structural image corruption is common, such as studies of children or cognitively impaired persons".

My specific comments are below:

- Same relaxometry value (table 1) is very simple. These values will most likely not match in children and subjects with cognitive impairment. Please comment and, if possible, include some results in children.

- Because of the simple tissue model and that fact that "specific care was taken to avoid the inclusion of partial volume voxels within the ROIs" the contrast comparison (eq 4) does not sufficiently identify quality of synthetic images. Probably a difference map between in-vivo and synthetic image would be more meaningful (and compare how it look wrt difference map between in-vivo repeats). Also, in table 2, In-vivo Hospital and Synthetic Hospital differ a lot for WM/GM.

- In all results, synthetic images are generated from same diffusion data. This does not provide evidence that the proposed method is robust to nuisances, motion and artifacts in diffusion data. It would have been specially interesting to see how it works when diffusion data is also corrupted by motion because subjects with motion in T1w/T2w images generally also have motion in diffusion. Also, its important to show repeatability of the method on diffusion data repeats.

- Because same diffusion data is used for all results, it is expected that the results will have very similar tracks and FA values. The source of the difference are small changes in ROI seeding, which will have minor impact on the tracks. So, in my humble opinion, these results (fig 4-5) are expected because same diffusion data was used.

- As the synthetic images are the only difference between these results (fig 4-5) - it is probably better to show dice and other metrics on freesurfer parcels (like thalamus etc.) than on binarised tractograms.

- Line 256: if I understand correctly, the statistical testing is incorrectly used in these results. A failure to reject the null hypothesis in a significance test does not mean that the null hypothesis is true. Wilcoxon signed-rank test tests the null hypothesis that two related paired samples come from the same distribution. A large p-value implies that null hypothesis can not be rejected and so we can not conclude anything.

- Line 294: "... accurate grey-matter/white-matter interface, which was generally of high quality here (Figure 3)." However the image show several differences in GM/WM boundaries. Please show difference map between synthetic and in-vivo image to justify this statement.

- Line 175: While processing with in-vivo structural images affine registration was used to co-register T1 and diffusion images (FA/b=0). It is well known that EPI distortion will be not resolved by affine registration. This might be the main issue in fig 5 (line 256-258).

- HCP-S dataset was chosen to have b-value of 3000s/mm2. Any reasons? It is related to use of CSD? Can results be included with b-value of 1000s/mm2, which is far more common.

- There are some similarity of the presented method to following works. Authors should consider highlighting differences:

Bhushan C et al. "Co-registration and distortion correction of diffusion and anatomical images based on inverse contrast normalization." NeuroImage. 2015;115:269–280.

Roy, Snehashis, et al. "Patch based synthesis of whole head MR images: Application to EPI distortion correction." International Workshop on Simulation and Synthesis in Medical Imaging. Springer, Cham, 2016.

Minor points:

- "... structural images have been corrupted." However approach does not fix corruputed images but just replaces them. It abstract and introduction gives an impression that corrupted images are fixed. It should be clarified.

- DWIs are also structural images. https://jnnp.bmj.com/content/75/9/1235

6. PLOS authors have the option to publish the peer review history of their article (what does this mean?). If published, this will include your full peer review and any attached files.

Reviewer #1: No

Reviewer #2: No

Reviewer #3: No

---

## [Author Response · Author response to Decision Letter 0]

6 Sep 2021

Please see the response to the reviewers in the document attached.

---

## [Decision Letter · Decision Letter 1]

8 Oct 2021

PONE-D-21-03790R1Avoiding Data Loss: Synthetic MRIs Generated from Diffusion Imaging Can Replace Corrupted Structural Acquisitions For Freesurfer-Seeded TractographyPLOS ONE

Dear Dr. Beaumont,

Thank you for submitting your manuscript to PLOS ONE. After careful consideration, we feel that it has merit but does not fully meet PLOS ONE’s publication criteria as it currently stands. Therefore, we invite you to submit a revised version of the manuscript that addresses the points raised during the review process. In particular, Reviewer 3 requested some clarifications about the goal of the algorithm and the statistical analysis. Please ensure that your decision is justified on PLOS ONE’s publication criteria and not, for example, on novelty or perceived impact.

We look forward to receiving your revised manuscript.

Kind regards,

Dzung Pham

Academic Editor

PLOS ONE

Journal Requirements:

Reviewers' comments:

Reviewer's Responses to Questions

**Comments to the Author**

1. If the authors have adequately addressed your comments raised in a previous round of review and you feel that this manuscript is now acceptable for publication, you may indicate that here to bypass the “Comments to the Author” section, enter your conflict of interest statement in the “Confidential to Editor” section, and submit your "Accept" recommendation.

Reviewer #2: All comments have been addressed

Reviewer #3: (No Response)

2. Is the manuscript technically sound, and do the data support the conclusions?

Reviewer #2: Yes

Reviewer #3: Yes

3. Has the statistical analysis been performed appropriately and rigorously? 

Reviewer #2: Yes

Reviewer #3: Yes

4. Have the authors made all data underlying the findings in their manuscript fully available?

Reviewer #2: Yes

Reviewer #3: Yes

5. Is the manuscript presented in an intelligible fashion and written in standard English?

Reviewer #2: Yes

Reviewer #3: Yes

6. Review Comments to the Author

Reviewer #2: (No Response)

Reviewer #3: I thank authors for updating the manuscript and clarifying several issues in the last version. In my humble opinion following clarifications before publication will be valuable to the readers:

- Based on the authors' responses and edits, it seems the intent and the ultimate goal of the proposed method is to obtain a synthetic T1w image that allows using existing DWI processing pipelines (like freesurfer) when an in-vivo T1w is absent. If so, I would suggest to further clarify this in abstract and introduction (Edits clarify it better in newer version, but can be further clarified). The earlier version gave impression that authors are suggesting a method that can replace in-vivo T1w/T2w images for all purposes.

- By “same diffusion data” comment, I meant that no repeatability results are shown for same subject. With explanation on SS3T and MSMT diffusion models, that comment is addressed.

- Can some statistical tests be added for results shown in table 2 and 3? This will help further strengthen the proposed method.

- Line 363-365: Authors mention statistical results for HCP-S dataset (in-vivo vs synthetic ). How about statistical tests of HCP-M dataset? Median of 0.59 vs 0.75 seems to imply substantial differences.

- Also, please report/add the statistical test results in table 4 and supplemental material as well.

- Please report actual p-values of each statistical tests instead of just mentioning "P>0.05". This is widely accepted as good scientific practice. https://doi.org/10.1007/s10654-016-0149-3

7. PLOS authors have the option to publish the peer review history of their article (what does this mean?). If published, this will include your full peer review and any attached files.

Reviewer #2: No

Reviewer #3: No

---

## [Author Response · Author response to Decision Letter 1]

19 Nov 2021

Please find the response to the editor and reviewers comments in the document attached to this submission.

---

## [Editor Report · Decision Letter 2]

1 Dec 2021

Avoiding Data Loss: Synthetic MRIs Generated from Diffusion Imaging Can Replace Corrupted Structural Acquisitions For Freesurfer-Seeded Tractography

PONE-D-21-03790R2

Dear Dr. Beaumont,

We’re pleased to inform you that your manuscript has been judged scientifically suitable for publication and will be formally accepted for publication once it meets all outstanding technical requirements.

Kind regards,

Dzung Pham

Academic Editor

PLOS ONE
---

## [Editor Report · Acceptance letter]

9 Feb 2022

PONE-D-21-03790R2 

Avoiding Data Loss: Synthetic MRIs Generated from Diffusion Imaging Can Replace Corrupted Structural Acquisitions For Freesurfer-Seeded Tractography 

Dear Dr. Beaumont:

I'm pleased to inform you that your manuscript has been deemed suitable for publication in PLOS ONE. Congratulations! Your manuscript is now with our production department. 

Kind regards, 

on behalf of

Dr Dzung Pham 

Academic Editor

PLOS ONE